# Structural basis of the activation of type 1 insulin-like growth factor receptor

Jie Li [1,4], Eunhee Choi [2,4]*, Hongtao Yu [2]* & Xiao-chen Bai [1,3]*

Type 1 insulin-like growth factor receptor (IGF1R) is a receptor tyrosine kinase that regulates cell growth and proliferation, and can be activated by IGF1, IGF2, and insulin. Here, we report the cryo-EM structure of full-length IGF1R–IGF1 complex in the active state. This structure reveals that only one IGF1 molecule binds the Γ-shaped asymmetric IGF1R dimer. The IGF1-binding site is formed by the L1 and CR domains of one IGF1R protomer and the α-CT and FnIII-1 domains of the other. The liganded α-CT forms a rigid beam-like structure with the unliganded α-CT, which hinders the conformational change of the unliganded α-CT required for binding of a second IGF1 molecule. We further identify an L1–FnIII-2 interaction that mediates the dimerization of membrane-proximal domains of IGF1R. This interaction is required for optimal receptor activation. Our study identifies a source of the negative cooperativity in IGF1 binding to IGF1R and reveals the structural basis of IGF1R activation.

[1] Department of Biophysics, University of Texas Southwestern Medical Center, Dallas, TX, USA. [2] Howard Hughes Medical Institute, Department of Pharmacology, University of Texas Southwestern Medical Center, Dallas, TX, USA. [3] Department of Cell Biology, University of Texas Southwestern Medical Center, Dallas, TX, USA. [4] These authors contributed equally: Jie Li, Eunhee Choi. *email: Eunhee.Choi@UTSouthwestern.edu; Hongtao.Yu@UTSouthwestern.edu; Xiaochen.Bai@UTSouthwestern.edu

The type 1 insulin-like growth factor receptor (IGF1R) is a class II receptor tyrosine kinase (RTK), belonging to the insulin receptor family, that plays critical roles in cell growth and differentiation[1,2]. Dysregulation of IGF1R has been implicated in human diseases, including both growth retardation and cancers[3,4]. In addition, IGF1 signaling affects the ageing process. Mutations of *Daf-2*, a gene encoding IGF1R in *C. elegans*, could lead to two times longer lifespan[5]. IGF1R and insulin receptor (IR) are closely related, and share 57% sequence identity and high structural similarity. Each IGF1R and IR protomer consists of the L1, CR, L2, FnIII-1,−2,−3, transmembrane (TM), and kinase domains (Supplementary Fig. 1a). Two such protomers are linked by multiple disulfide bonds, forming a stable, covalent dimer (Supplementary Fig. 1a). For clarity, the domains in protomers 1 and 2 are denoted as L1 – FnIII-3, and L1′ – FnIII-3′, respectively, in the following text.

The structure of the intact IGF1R ectodomain in apo, ligand-free state was determined by X-ray crystallography, showing a Λ-shaped dimer[6,7]. In this inactive dimeric structure, the two membrane-proximal FnIII-3 domains of IGF1R are spatially separated by ~67 Å, preventing the kinase domains from trans-autophosphorylation. This conformation is maintained in part by autoinhibitory inter-subunit interactions between L1 and FnIII-2′ in the dimer, and deletion of L1 leads to spontaneous activation of IGF1R. Previous FRET analyses suggest a model in which IGF1 releases the autoinhibition and induces large structural rearrangements of IGF1R to bring the two intracellular kinase domains into proximity for receptor activation[8]. This release-of-autoinhibition model for IGF1R activation is supported by a crystal structure of the IGF1R ectodomain in complex with IGF1, showing that IGF1 binding to IGF1R breaks the autoinhibitory interactions between the L1 and FnIII-2′ domains in the apo state[7]. This structure of the IGF1R–IGF complex likely represents an intermedia state during IGF1R activation, however, as the two FnIII-3 domains (and presumably the kinase domains connected to them) remain separated by a long distance. In addition, this structure does not provide an explanation for the negative cooperativity in the binding of IGF1 to IGF1R, a classic feature of both IR and IGF1R for which the mechanisms are poorly understood[9,10]. The lack of structures of the full-length, IGF1-bound IGF1R in the active conformation has hindered the understanding of the activation mechanism of IGF1R that involves negative cooperativity.

In this study, we determine the cryo-EM structure of the full-length IGF1R–IGF1 complex in the active state at an overall resolution of 4.3 Å. Our structure reveals a Γ-shaped asymmetric IGF1R dimer bound to only one IGF1 molecule. The IGF1-binding site is formed by the L1 and CR domains (CRD) of one IGF1R protomer, and the α-CT′ and FnIII-1′ domains of the other. The interface between the IGF1 C-domain loop and the IGF1R CR domain is revealed in detail. In addition, we show that, the liganded α-CT′ forms a rigid beam-like structure with the unliganded α-CT, which prevents its rearrangement required for the binding of a second IGF1 molecule. The rigid connection between the two α-CTs in the active, asymmetric IGF1R dimer thus explains the negative cooperativity in binding of IGF1 to IGF1R[9,10]. Finally, we identify an intra-protomer interaction, located close to the membrane, between the unliganded L1′ and FnIII-2′ domains. This interaction plays an important role in receptor activation through stabilizing the active IGF1R dimer. Taken together, our results indicate that binding of one IGF1 molecule is sufficient to break the autoinhibited state of IGF1R, leading to receptor activation.

## Results

**Overall structure of the full-length IGF1R–IGF1 complex**. We successfully reconstituted the complex of full-length mouse IGF1R (MmIGF1R) and human IGF1 (HsIGF1) in vitro for cryo-EM analysis (Supplementary Fig. 1b). We used MmIGF1R (which shares 96% sequence identity with HsIGF1R) for cryo-EM structural determination, as it had higher expression level than did HsIGF1R. Several key residues in the IGF1R C-terminal tail involved in receptor internalization were mutated to further increase the expression yield[11–13]. IGF1 moderately increased autophosphorylation of the recombinant IGF1R isolated with detergent, although IGF1R exhibited high background activity even in the absence of IGF1 (Supplementary Fig. 1c, d). The 3D reconstruction of the MmIGF1R–HsIGF1 complex (referred to simply as IGF1R–IGF1 hereafter) was determined at 4.3 Å resolution (Supplementary Figs. 2–4, Table 1). We built a nearly complete model for the ectodomain of the IGF1R dimer and one IGF1 molecule, with the help of the crystal structures of IGF1R and IGF1[7] (Fig. 1a and Supplementary Fig. 4). No intra-domain conformational changes were observed at this level of resolution. The transmembrane (TM) domain was determined only to low resolution, presumably due to structural flexibility (Supplementary Fig. 4). Although we attempted extensive focused classification, the densities of kinase domains were still not resolved in the cryo-EM map. We will use the amino acid numbering of human IGF1R to describe our subsequent structural and functional analyses, as most previous studies reported in the literature focus on human IGF1R.

The overall structure of the IGF1R–IGF1 complex has an asymmetric Γ shape (Fig. 1a, b). The top part of the Γ is composed of L2 and FnIII-1 domains from both protomers as well as L1 from protomer 1; while the lower part of the Γ consists of FnIII-2 and FnIII-3 from both protomers as well as L1′ from protomer 2 (Fig. 1b). Only one IGF1 molecule is bound to the IGF1R dimer; it is located at the top part of the Γ (Fig. 1a, b). This 1:1 stoichiometry of IGF1:IGF1R dimer is consistent with previous results showing that binding of one IGF1 molecule to the IGF1R dimer hinders the binding of a second IGF1 (i.e. negative cooperativity)[9,10]. The distance between the two membrane-proximal FnIII-3 domains in the IGF1-bound IGF1R is ~39 Å, which is much shorter than that in the apo IGF1R dimer (~65 Å). The dimerized TM conformation further indicates that the two intracellular kinase domains are positioned in proximity for trans-autophosphorylation (Supplementary Fig. 4). These structural features suggest that this conformation of the IGF1R dimer represents the active state.

Our structure of the IGF1R–IGF1 complex, obtained using the full-length IGF1R, show some similarities to the recently reported negative stain EM result of the full-length IR[14] as well as the cryo-EM structure of an engineered insulin receptor (IR) ectodomain bound to insulin[15]. Both cryo-EM structures of IGF1R and IR exhibit an asymmetric Γ shape with only one ligand molecule bound. There are important differences between the two structures, however. For example, the distance between the two FnIII-3 domains in the structure of the IR ectodomain–insulin complex (~16 Å) is much shorter than that in our IGF1R–IGF1 structure (Supplementary Fig. 5a). We note that a leucine zipper motif is fused to the FnIII-3 domain of the IR ectodomain, which is designed to stabilize the active conformation of the engineered IR. Therefore, the structural differences observed here could be due to the different sample preparation methods and may not reflect true differences between the active conformations of these receptors.

**Table 1 Cryo-EM data collection, refinement, and validation statistics**

| | IGF1R-IGF1 complex (EMDB-20524) (PDB 6PYH) |
|---|---|
| **Data collection and processing** | |
| Magnification | 46,730 |
| Voltage (kV) | 300 |
| Electron exposure (e–/Å²) | 50 |
| Defocus range (μm) | 1.5–3 |
| Pixel size (Å) | 1.07 |
| Symmetry imposed | C1 |
| Initial particle images (no.) | 1,431,211 |
| Final particle images (no.) | 291,978 |
| Map resolution (Å) | 4.3 |
| FSC threshold | 0.143 |
| Map resolution range (Å) | 4.3– |
| **Refinement** | |
| Initial model used (PDB code) | 5U8Q |
| Model resolution (Å) | 4.3 |
| FSC threshold | 0.143 |
| Model resolution range (Å) | 4.3 |
| Map sharpening B factor (Å²) | −110 |
| **Model composition** | |
| Non-hydrogen atoms | 13187 |
| Protein residues | 1648 |
| Ligands | |
| **B factors (Å²)** | |
| Protein | −110 |
| Ligand | |
| **R.m.s. deviations** | |
| Bond lengths (Å) | 0.007 |
| Bond angles (°) | 0.876 |
| **Validation** | |
| MolProbity score | 2.65 |
| Clashscore | 28.12 |
| Poor rotamers (%) | 0.55 |
| **Ramachandran plot** | |
| Favored (%) | 81.65 |
| Allowed (%) | 18.16 |
| Disallowed (%) | 0.37 |

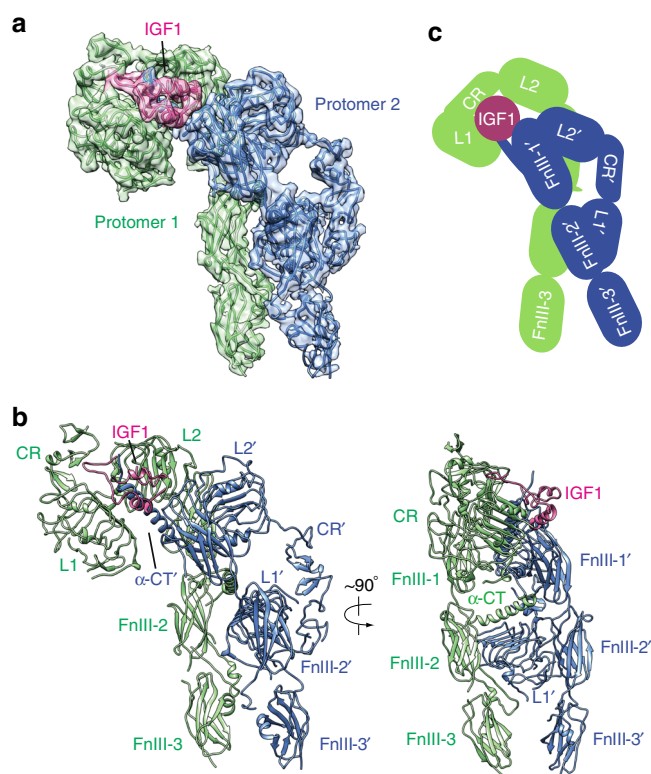

**Fig. 1** Overall structure of the IGF1R-IGF1 complex. **a** 3D reconstruction of the IGF1R dimer with one IGF1 bound and the corresponding ribbon representation of this complex fitted into cryo-EM map. **b** The ribbon representation of IGF1R-IGF1 complex showing in two orthogonal views. **c** Cartoon representation of the IGF1R-IGF1 complex depicting the IGF1R dimer bound with one IGF1. The two IGF1R protomers are colored in green and blue, respectively; the IGF1 is colored in pink. The name of each domain is labeled as follows: L1 and L2 (leucine-rich repeat domains); CR (cysteine-rich domain); FnIII-1, FnIII-2 and FnIII-3 (fibronectin type III domains); α-CT (the C-terminus of the α-subunit)

**Structural transitions of IGF1R induced by IGF1.** Significant conformational rearrangements are observed in the IGF1R–IGF1 complex with respect to the apo IGF1R. In the Λ-shaped IGF1R apo dimer, each leg of the Λ that comprises FnIII-1, −2, and −3 domains are stabilized by the interactions between L1 and FnIII-2′ domains from the two protomers in the dimer (Fig. 2a). Binding of IGF1 to the L1/α-CT′ site of IGF1R, the primary IGF1-binding site defined previously by X-ray crystallography[7], breaks one of the two equivalent L1–FnIII-2′ interactions in the apo state, thus weakening this autoinhibited conformation. The remaining L1′–FnIII-2 interaction maintains the unliganded half of the IGF1R dimer in the apo state, resulting in the asymmetric architecture. The L1 domain and α-CT′ of the liganded half of IGF1R, together with the bound IGF1, undergo a hinge motion and move upward to form the top part of the Γ (Fig. 2b). The FnIII-1′, −2′, and −3′ domains of protomer 2 swing toward to the other leg, and make close contact with the L1′ domain of the same protomer (Fig. 2b). This swing motion of FnIII-1′, −2′, and −3′ domains upon ligand binding reduces the distance between the two legs, thus facilitating the autophosphorylation of intracellular kinase domains (Fig. 2c). Notably, similar structural transitions also occur during the insulin-induced activation of IR (Supplementary Fig. 5b), suggesting that the mechanism underlying activation may be conserved between these two closely related receptors.

To test whether binding of one IGF1 molecule to the IGF1R dimer is sufficient for receptor activation, we transfected HEK293 cells with the IGF1R WT plasmid alone or with a 1:1 mixture of the IGF1R WT plasmid and a plasmid that encodes IGF1R F701A, a mutant deficient in IGF1 binding. Co-expression of IGF1R WT and F701A at equal levels is expected to generate 50% WT:F701A heterodimer, 25% WT:WT homodimer, and 25% F701A:F701A homodimer. If binding of one IGF1 molecule to the IGF1R dimer is sufficient for receptor activation, then both the WT:WT and WT:F701A dimers are expected to be active, and the IGF1R activity in cells transfected with WT and F701A plasmids will be 75% of that in cells transfected with the WT plasmid alone. If binding of two IGF1 molecules to the IGF1R dimer is required for receptor activation, the IGF1R activity in cells transfected with WT and F701A plasmids should be 25% of that in cells transfected with the WT plasmid alone. Our cellular assays showed that the IGF1R activity in cells with the co-expression of WT and F701A was ~75% of that in cells expressing only the WT receptor, indicating that the WT:F701A heterodimer was indeed active (Fig. 2d, e). To estimate the relative expression of IGF1R WT and F701A in cells, we transfected Myc-tagged IGF1R WT alone or the 1:1 mixture of Myc-tagged IGF1R WT and untagged IGF1R F701A into cells. IGF1R WT and F701A were expressed at similar levels, and the F701A activity in cells with the co-expression of WT and F701A was similar to WT (Fig. 2f, g). This result is consistent with the previous finding that

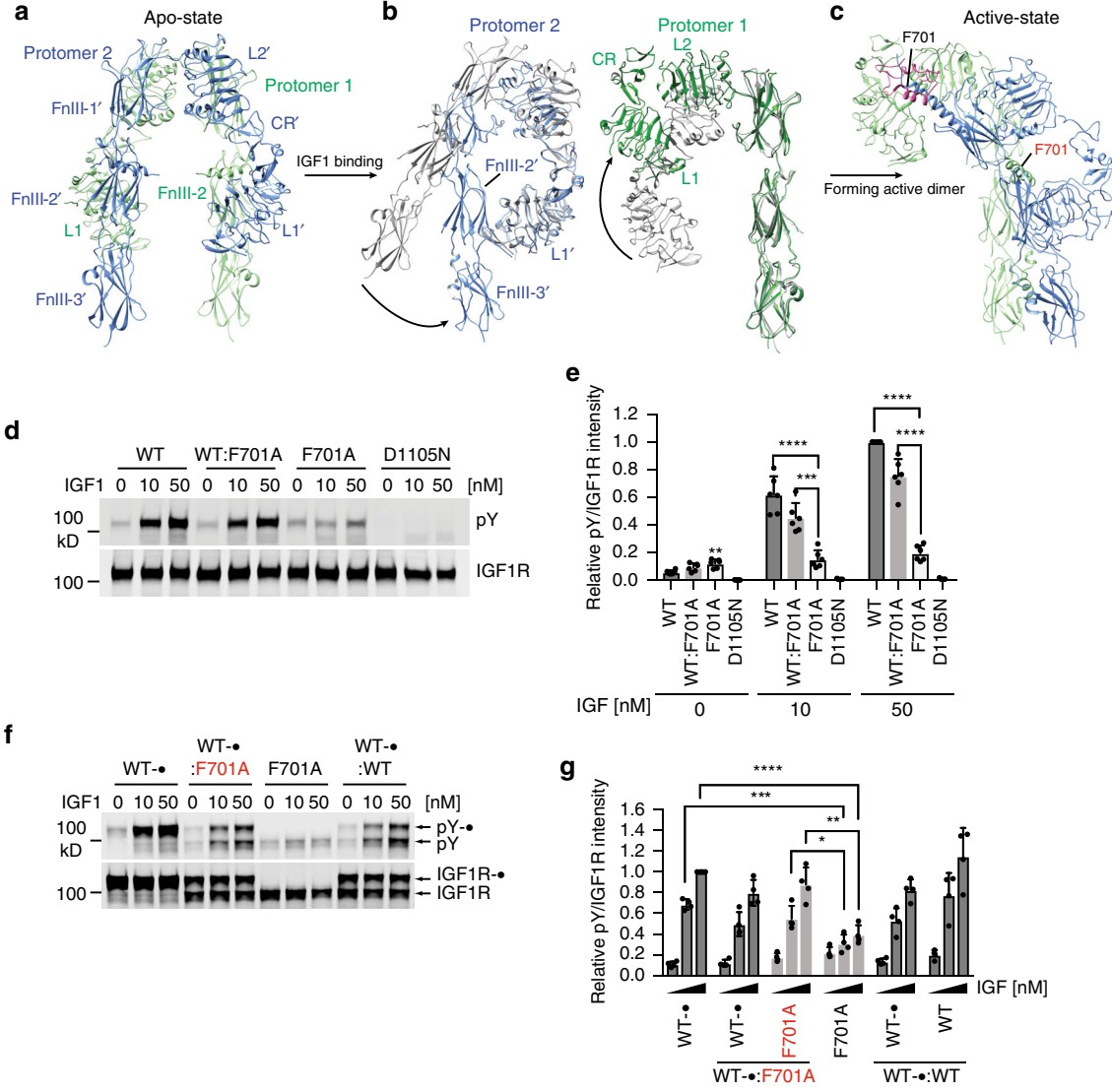

**Fig. 2** Conformational change of each IGF1R protomer upon IGF1 binding. **a** Overall view of apo-IGF1R dimer. **b** Superposition between apo IGF1R protomer (gray) and the two different IGF1R protomers in the active state (blue or green) separately, revealing two different types of conformational changes upon IGF1 binding (indicated by the arrows). **c** Overall view of active-IGF1R dimer. **d** IGF1-induced IGF1R autophosphorylation in 239FT cells expressing wild-type (WT) alone, WT:F701A or F701A alone. Kinase dead mutant (D1105N) was used as a negative control. **e** Quantification of the western blot data shown in **d** (Mean ± SD). Each experiment was repeated six times. Significance calculated using two-tailed students *t*-test; between WT and mutant; **$p < 0.01$ ***$p < 0.001$, and ****$p < 0.0001$. **f** IGF1-induced IGF1R autophosphorylation in 293FT cells expressing Myc-tagged WT (WT-●) alone, Myc-tagged WT: untagged F701A (WT-●:F701A), untagged F701A alone or Myc-tagged WT: untagged WT (WT-●:WT). **g** Quantification of the western blot data shown in **f** (Mean ± SD). Each experiment was repeated three times. Significance calculated using two-tailed students *t*-test; *$p < 0.05$, **$p < 0.01$, ***$p < 0.001$, and ****$p < 0.0001$. Source data are provided as a Source Data file

WT:WT and WT:F701A dimers exhibit similar IGF1-binding affinities[16], and suggest that binding of only one IGF1 molecule to the IGF1R dimer is sufficient for receptor activation.

**The IGF1-binding site in the active state of IGF1R**. IGF1 binding to L1/α-CT′ of IGF1R disrupts the constraint that is responsible for autoinhibition[8], thereby allowing the structural rearrangement from the Λ-shaped inactive dimer to the Γ-shaped active dimer. IGF1 also stabilizes the active IGF1R dimer by simultaneously contacting several different domains between the two protomers of IGF1R (Fig. 1a, c). In such a way, IGF1 crosslinks the two protomers in the active dimer.

The primary IGF1-binding site observed in our IGF1R-IGF1 cryo-EM structure consists of the L1 domain and α-CT′ (Fig. 3a),

and is nearly identical to that in the crystal structure of the IGF1R ectodomain bound to IGF1[7]. Therefore, we will not describe this site in detail here. The residues comprising the primary site are highly conserved between IGF1R and IR with one exception (Fig. 3i). At the primary site of IR, F39 and K40 of IR pack against Y B16 and E B21 of insulin, but this interaction does not exist in the IGF1R–IGF1 complex.

In addition to this primary site, the same IGF1 contacts a secondary sub-site in the active IGF1R dimer, which involves a loop on the top part of the FnIII-1′ domain (Fig. 3a). A similar secondary sub-site was first observed in the cryo-EM structure of the IR–insulin complex[15,17] (Fig. 3i). However, the residues involved in this secondary ligand-binding site are not highly conserved between IGF1R and IR. Likewise, residues in IGF1 and insulin that contact this site are not very conserved. At this

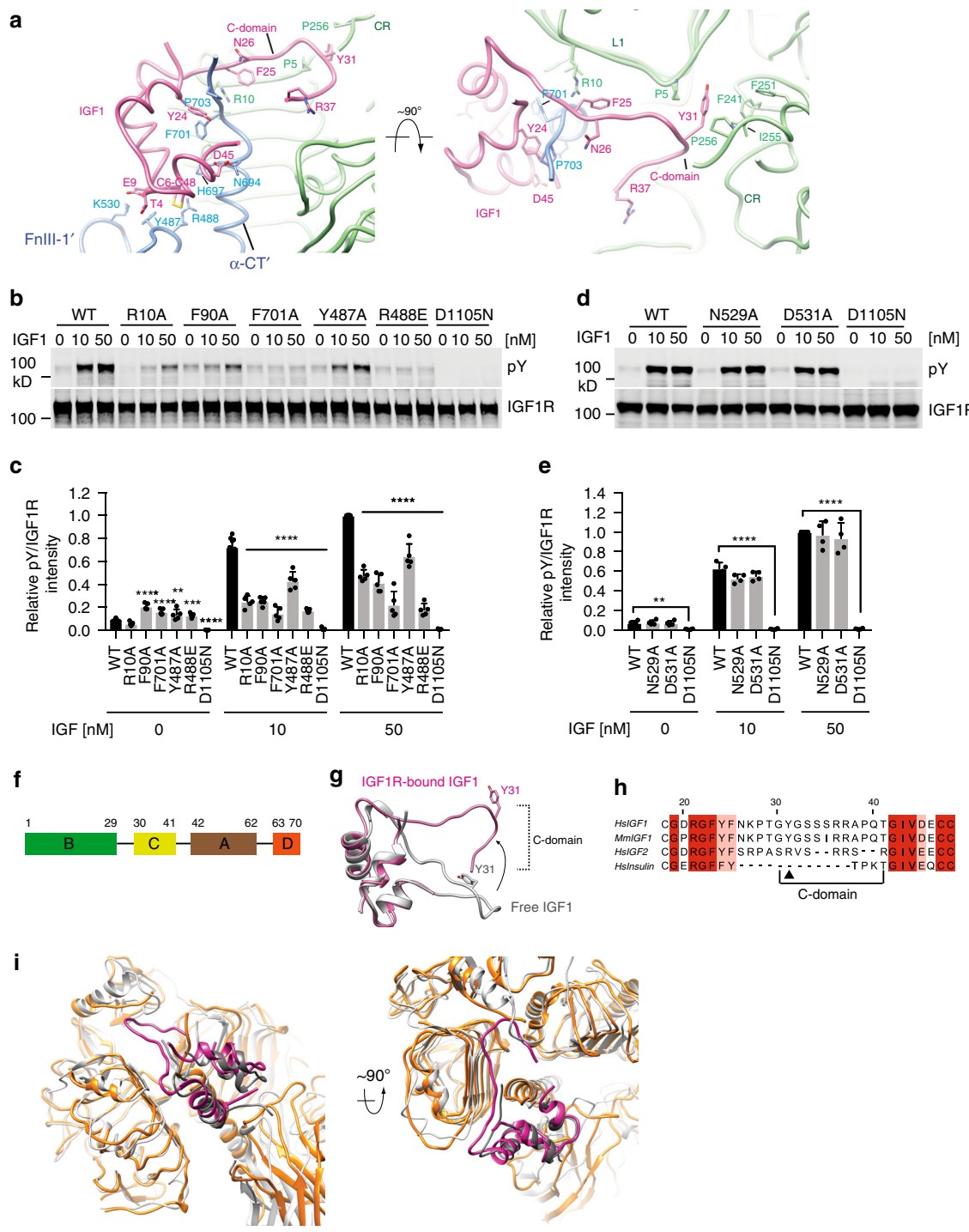

**Fig. 3** IGF1 binding site in the active-IGF1R dimer. **a** Close-up view of the interaction between IGF1R and IGF1 in two orthogonal views. **b** IGF1-induced IGF1R autophosphorylation in 293FT cells expressing WT IGF1R or indicated mutants. The kinase dead mutant (D1105N) was used as a negative control. **c** Quantification of the western blot data shown in **b** (Mean ± SD). Each experiment was repeated five times. Significance calculated using two-tailed students t-test; between WT and mutants; **p < 0.01, ***p < 0.001, and ****p < 0.0001. **d** IGF1-induced IGF1R autophosphorylation in 293FT cells expressing WT IGF1R or indicated mutants. **e** Quantification of the western blot data shown in **d** (Mean ± SD). Each experiment was repeated four times. Significance calculated using two-tailed students t-test; **p < 0.01 and ***p < 0.001. **f** Domain organization of IGF1. **g** Superposition between apo-IGF1 (gray) and IGF1 after binding IGF1R (pink), revealing the conformational change of C-domain (indicated by the arrow). **h** Sequence alignment of human IGF1, mouse IGF1, human IGF2, and insulin, showing the non-conserved C-domain. **i** Superposition between IGF1-bound IGF1R (pink and orange) and Insulin-bound IR (dark gray and light gray). Source data are provided as a Source Data file

secondary IGF1-binding sub-site, Y487 and R488 of IGF1R pack closely against residues T4, C6, and E9 of IGF1 (Fig. 2a), of which E9 has previously been shown to be important for receptor binding[18]. In insulin, the residue corresponding to T4 of IGF1 is changed to a histidine (H B5), which packs against P495 and F497 of IR at the secondary insulin-binding site. To validate the functional importance of the secondary IGF1-binding site, we introduced Y487A and R488A mutations into IGF1R, and tested their effects on IGF1R activation. Indeed, both mutants showed significantly reduced IGF1-dependent IGF1R activation in a wide range of IGF1 concentrations and at different time points (Fig. 3b, c, Supplementary Fig. 6); whereas mutation of N529 and D531, which were located at the periphery of this site, had no effect (Fig. 3d, e).

Replacing the C-domain loop of IGF1 with a short GGGG loop completely abolishes its binding to IGF1R[19] (Fig. 3f). The C-domain loop of IGF1 is thus important for receptor binding, but how it does so is unknown. In our structure of the IGF1R–IGF1 complex, the C-domain loop of IGF1 inserts into a groove formed by L1 and CRDs of IGF1R (Fig. 3a). Due to the lack of side chain densities, we cannot accurately model side chain conformations for most of residues in this loop, except for Y31 whose side chain is clearly resolved in the cryo-EM map (Supplementary Fig. 4). The aromatic side chain of Y31 of IGF1 is sandwiched between P5 in the L1 domain and P256 in the CRD, and is further surrounded by several other conserved hydrophobic residues in the CRD, including F241, F251, and I255 (Fig. 3a). Moreover, superimposition of the model of receptor-bound IGF1 onto that of free IGF1 determined by NMR (PDB: 1PMX)[20] revealed that a large relocation of the C-domain loop occurs upon IGF1 binding to IGF1R (Fig. 3g). These interactions provide a structural basis for the previous mutagenesis results of IGF1 showing that the Y31A mutation of IGF1 reduces its receptor-binding affinity by six fold[21]. The F241A or F251A mutations in the CRD of IGF1R also dramatically reduce the binding of IGF1 to IGF1R, further validating this IGF1-binding interface[22]. Notably, this C-domain loop is shorter in IGF2 or insulin (Fig. 3h, i), explaining why the binding of IGF2 or insulin to IGF1R is much weaker than that of IGF1[23,24].

In the active state of IGF1R, IGF1 simultaneously engages multiple domains and motifs from both protomers of IGF1R, including the L1, CR, α-CT′, and FnIII-1′ domains. This ligand-binding mode supports the previous hypothesis that the activation of IGF1R requires the cross-linking of two protomers in a particular conformation by IGF1[25]. Indeed, disruption of any of these interfaces leads to the deficiency in IGF1-induced IGF1R activation. Together, these results suggest that IGF1 activates IGF1R through disrupting the autoinhibited apo state and stabilizing the active state of IGF1R.

**Structural basis of the negative cooperativity**. Our structural model of the IGF1R–IGF1 complex provides key mechanistic insight into the negative cooperativity in the binding of IGF1 to IGF1R. It has long been known that the binding of IGF1 is accompanied by a large-distance relocation of the α-CT motif with respect to the L1 domain (Fig. 4a)[7], suggesting that certain structural flexibility of α-CT is required for ligand binding. In the apo structure of IGF1R, only the N-terminal portion of α-CT (residues 682–704) adopts a short α-helix, while the rest of this motif is disordered[7]. This partially folded α-CT does not have any structural constraints in either N- or C-termini, thus allowing the conformational change of α-CT for IGF1 binding (Fig. 4a). The two equivalent binding sites in the apo dimer are identical, suggesting that IGF1 can bind to either of the two sites in the apo state with the same probability.

Binding of IGF1 to either of the two equivalent binding sites in IGF1R triggers the structural rearrangement in half of the apo IGF1R dimer. As a result, the L1 domain, α-CT′, as well as the bound IGF1 move upward to the top part of the IGF1R dimer (Fig. 2b). Along with this conformational change, residues 670–681 in the N-terminal portion of the liganded α-CT′, which is disordered in the apo state, become a well ordered α-helix that contacts the upper part of FnIII-1′ (Fig. 4b–d). Consequently, the liganded α-CT′ in the active state of IGF1R form a much longer α-helix, as compared to that in the apo state (Fig. 4b, c). In addition, several residues (C670–T675) that link the N-terminus of this long α-CT′ helix to the N-terminus of the unliganded α-CT interact with the FnIII-1′ domain, and serve as an anchor to couple the two α-CTs (Fig. 4d).

As the two α-CT motifs in IGF1R dimer is covalently linked by multiple disulfide bonds between three cysteine residues C669, C670, and C672, the rigidification and relocation of the α-CT′ induced by binding of one IGF1 leads to the conformational change of the other α-CT in the adjacent protomer (Fig. 4b–e). Residues 670–675 in the N-terminal portion of the unliganded α-CT is dragged upward by the liganded α-CT′, and make close contact with the lower part of FnIII-1′ (Fig. 4c, d). This interaction, in turn, drives the entire unliganded α-CT to form a rigid long α-helix, similar to that formed by the liganded α-CT′ (Fig. 4b). Furthermore, the interaction between the unliganded α-CT and the FnIII-1′ domain, together with the disulfide bonds between the two α-CTs, restricts the conformational change of unliganded α-CT, which is required for the binding of a second IGF1 molecule (Fig. 4c, e). Therefore, this unliganded α-CT in the asymmetric IGF1R dimer bound to one IGF1 is restrained in a position that disfavors the binding of a second IGF1. This structural feature explains the negative cooperativity in the binding of IGF1 to IGF1R.

The two α-CT motifs are structurally uncoupled in the apo state of IGF1R, allowing the efficient binding of IGF1 to either of the two primary sites. In contrast, two α-CT motifs in the active IGF1R dimer bound to one IGF1 are physically coupled because of the lengthening of the two α-CT helices and are tethered by disulfide bonds (Fig. 4b, c). We hypothesized that disrupting this tether or lengthening the linker that connects the two α-CTs may uncouple the two binding sites in the active state of IGF1R, reduce the negative cooperativity, and allow the binding of the second IGF1. Consistent with this hypothesis, deletion of the cysteines (C669, C670, and C672; Δ3 C) that formed the disulfides or the insertion of four glycine residues (P673G4) in the linker connecting the two α-CT motifs increased the binding of IGF1 to IGF1R by ~50% and reduced the negative cooperativity (Fig. 4f, g). Interestingly, these IGF1R mutants with lower negative cooperativity in IGF1 binding showed slightly weaker activation in response to IGF1 (Fig. 4h, i), suggesting that IGF1R with one IGF1 bound exhibits higher activity than that with two IGF1s bound. Therefore, IGF1R with one IGF1 bound is the most optimal state for activation, which partially explains the functional importance of negative cooperativity.

**Requirement of L1′–FnIII-2′ interaction for IGF1R activation**. In addition to the IGF1R–IGF1 interactions, we identified an intra-protomer interaction, which only exists in the active state of IGF1R. This interaction is located in the membrane-proximal region of IGF1R, involving the unliganded L1′ and FnIII-2′ domains from the same protomer (Fig. 5a). Specifically, the β hairpin motif (residues 163–174) of L1′ contacts a flat surface on FnIII-2′ that is formed by strands β1 and β7 (Fig. 5b). Many conserved residues in the L1′ domain, including K164, T165, T166, N169, Y171, and Y173, contribute to this interaction.

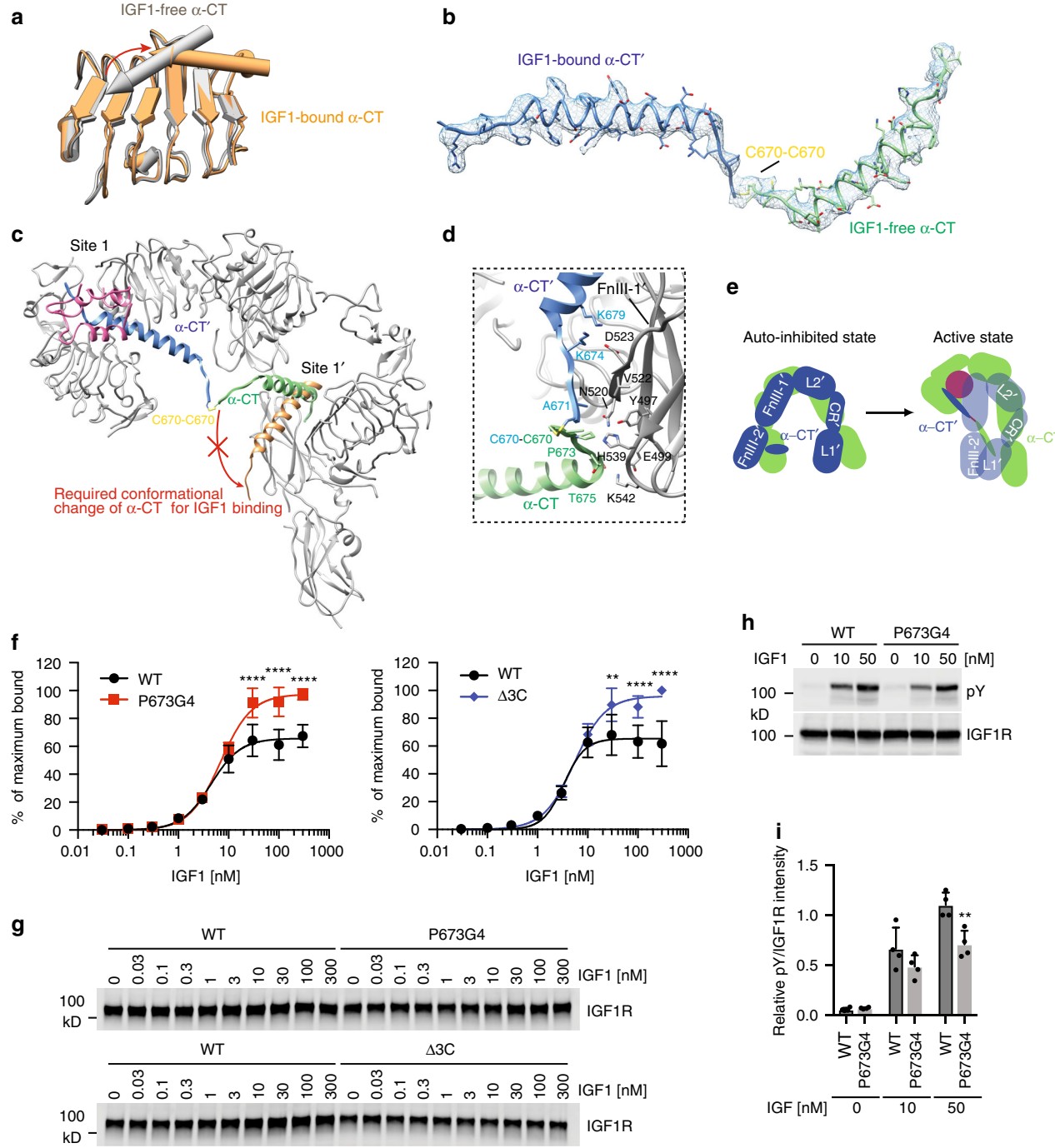

**Fig. 4** The source of negative cooperativity in the binding of IGF1 to IGF1R. **a** Superposition between IGF1 bound L1/α-CT′ (orange) and apo L1/α-CT′ (grey) by aligning the L1 domain, revealing the conformational change of α-CT′ upon IGF1 binding. **b** Cryo-EM density of dimerized α-CT motifs. **c** Overall view of dimerized α-CT motifs in the context of IGF1R active dimer. Part of the model is removed for clarity. IGF1 bound α-CT′ (orange) is superposed with the unliganded α-CT (green) in the IGF1R active dimer by aligning the L1 domain, revealing the required conformational change of α-CT for IGF1 binding (indicated by a red arrow). **d** Close-up view of the α-CTs/FnIII-1′ interaction in the IGF1R active dimer. **e** Cartoon model illustrating the structural difference of two α-CT motifs between the apo and active states of IGF1R. **f** Binding of IGF1 labeled with Alexa Fluor 488 to 293FT cells expressing IGF1R WT or IGF1R mutants (Mean ± SD). Each experiment was repeated three times. Significance calculated using two-tailed students t-test; between WT and mutants, $**p < 0.01$ and $****p < 0.0001$. **g** Representative western blot images of the amount of IGF1R in **f**. **h** IGF1-induced IGF1R autophosphorylation in 293FT cells expressing WT IGF1R or P673G4. **i** Quantification of the western blot data shown in **h** (Mean ± SD). Each experiment was repeated four times. Significance calculated using two-tailed students $t$-test; between WT and mutants; $**p < 0.01$. Source data are provided as a Source Data file

Besides this newly identified L1′–FnIII-2′ intra-protomer interaction, there is an L1′–FnIII-2 inter-protomer interaction that already exists in the apo state of IGF1R and persists in the active state. Through these two distinct types of interactions, the unliganded L1′ domain effectively bridges the two legs of the active

IGF1R dimer and brings them into close proximity for receptor activation (Fig. 5a).

To test whether this newly identified L1′–FnIII-2′ interaction is functionally important for receptor activation, we introduced multiple mutations in the L1 β hairpin of IGF1R (K164A, T166A,

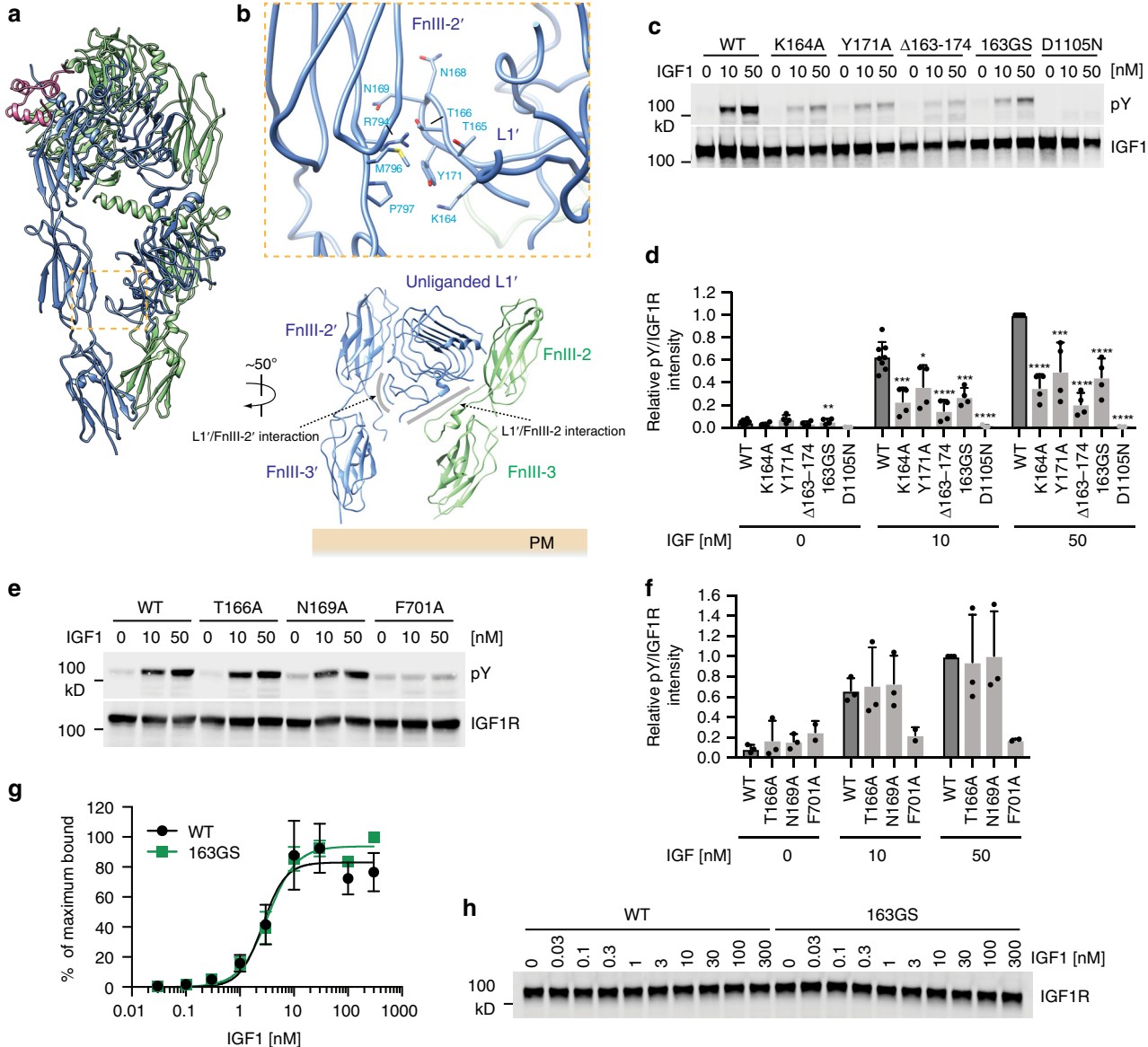

**Fig. 5** L1'/FnIII-2' interaction in the active-IGF1R dimer. **a** Overall view of IGF1R active dimer, showing in two different views. **b** Close-up view of the L1'/FnIII-2' interaction. The location of this interaction in the IGF1R active dimer is indicated by an orange box in **a**. **c** IGF1-induced IGF1R autophosphorylation in 293FT cells expressing WT IGF1R or indicated mutants. The kinase dead mutant (D1105N) was used as a negative control. **d** Quantification of the western blot data shown in **c** (Mean ± SD). Each experiment was repeated four times. Significance calculated using two-tailed students t-test; between WT and mutants; *$p < 0.05$, **$p < 0.01$, ***$p < 0.001$, and ****$p < 0.0001$. **e** IGF1-induced IGF1R autophosphorylation in 293FT cells expressing WT IGF1R or indicated mutants. The IGF1-binding mutant (F701A) was used as a negative control. **f** Quantification of the western blot data shown in **e** (Mean ± SD). Each experiment was repeated three times. **g** Binding of IGF1 labeled with Alexa Fluor 488 to 293FT cells expressing IGF1R WT or 163GS (Mean ± SD). Each experiment was repeated three times. **h** Representative western blot images of the amount of IGF1R in **g**. Source data are provided as a Source Data file

N169A, and Y171A), deleted the entire L1 β hairpin (Δ163–174), or replaced this L1 β hairpin with a short GSGS loop (163GS) to disrupt this interaction. Consistent with our structure-based predictions, most of these mutants were deficient in IGF1-dependent activation (Fig. 5c, d, Supplementary Fig. 6). However, T166A or N169A single mutation did not change the activation (Fig. 5e, f). Our in vitro IGF1-binding assays showed that IGF1R WT and 163GS bound to IGF1 with similar affinities, indicating that the L1 hairpin mutations did not affect IGF1 binding (Fig. 5g, h). Thus, the unliganded L1' domain plays an important role in receptor activation, through positioning the membrane-proximal stalk domains.

## Discussion

Our structural and functional analyses reveal that binding of one IGF1 molecule to IGF1R disrupts the Λ-shaped autoinhibited IGF1R dimer and triggers a conformational rearrangement that leads to the formation of an asymmetric Γ-shaped active dimer (Fig. 6). This active dimer is further stabilized by extensive contacts of the IGF1 molecule with multiple IGF1R domains in both protomers, namely L1, α-CT', FnIII-1', and CRD. The L1'–FnIII-2' intra-protomer interaction in the membrane-proximal region further stabilizes the active dimer. These structural rearrangements reduce the distance between the two intracellular kinase domains, and thus promote trans-autophosphorylation and activation.

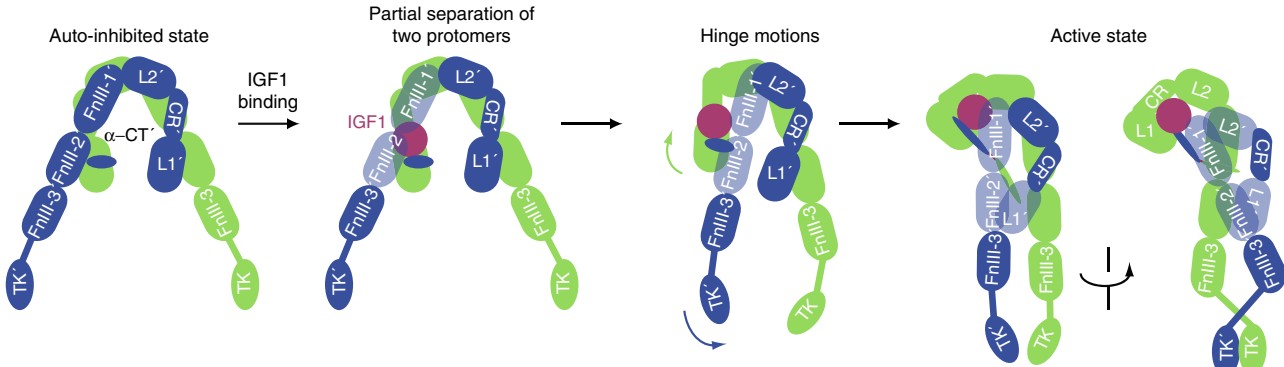

**Fig. 6** Cartoon representation of a working model for IGF1-induced IGF1R activation. The two IGF1R protomers are colored in green and blue, respectively; the IGF1 is colored in pink. The name of each domain is labeled as follows: L1 and L2 (leucine-rich repeat domains); CR (cysteine-rich domain); FnIII-1, FnIII-2, and FnIII-3 (fibronectin type III domains); α-CT (the C-terminus of the α-subunit); TK (tyrosine kinase domain)

Previous ligand binding assays have identified that the C-domain loop in IGF1 is essential for its binding to IGF1R[19,21]. Our structure reveals the role of the C-domain loop of IGF1 in receptor binding. In the active state of IGF1R, the C-domain loop of IGF1 adopts an extended loop that contacts both L1 and CRD of IGF1R. The C-domain loops in IGF2 and insulin are shorter and thus cannot reach the CRD, which explains why the CRD of IGF1R is only important for IGF1 binding, but not for IGF2 or insulin binding[24]. The two consecutive arginines (R36 and R37) in the C-domain loop of IGF1 have also been shown to play a role in receptor binding[26]. We cannot model these two residues unambiguously due to the lack of side chain densities. Nevertheless, based on clear cryo-EM density, we can determine the backbone positions of R36 and R37, which are placed close to the L2 domain of IGF1R. Therefore, it is likely that R36 and R37 of IGF1 contribute to the IGF1R–IGF1 interaction through contacting the L2 domain of IGF1R.

Negative cooperativity in ligand binding has long been known to be an intrinsic property for both IGF1R and IR. In this work, we provide the structural basis for the negative cooperativity of IGF1 binding to IGF1R. In the asymmetric active dimer of IGF1R with only one IGF1 bound, the two α-CT motifs from both protomers converge at the FnIII-1′ domain and form a rigid helical beam. This beam is stabilized by the disulfide bonds between the two α-CTs and the interaction between α-CT and FnIII-1′. Such a rigid beam-like structure hinders the conformational change of unliganded α-CT required for ligand binding, and hence prevents the binding of the second IGF1. Conversely, the second IGF1 binding to the low affinity site may disturb the first IGF1-bound site through the coupled α-CT motifs, leading to the accelerated dissociation of the already bound IGF1. In IR, enhancing the α-CT structural flexibility by the insertion of four glycines in the N-terminal part of α-CT increases insulin-binding affinity[15], suggesting that the origin of negative cooperativity between IR and IGF1R may be similar.

During the activation of IGF1R, the two L1 domains from two protomers (L1 and L1′) play distinct roles. One of the two L1s is bound to IGF1, and moves upward to the top part of the Γ after ligand binding, whereas the other L1 domain (L1′) behaves as a pseudo-ligand to bring the two legs of IGF1R into close proximity for receptor activation. Our 3D classification results show that the unliganded L1′ exhibits larger degrees of flexibility than the liganded L1. We speculate that the structural flexibility of L1′ is caused by the weak binding of a second IGF1 in a subset of IGF1R molecules. This may happen more frequently with increased amount of excess IGF1. In some 3D classes, the unliganded L1′ is slightly rotated away from the middle of two legs, and the

distance between the two legs becomes larger (Supplementary Fig. 3). This observation suggests that the displacement of L1′ by the weak binding of a second IGF1 may disrupt the functionally important L1′–FnIII-2′ interaction, which may lead to sub-optimal receptor activation. In support of this hypothesis, previous BRET experiments suggest that the distance between the two IGF1R intracellular domains increases in the presence of high concentrations of IGF1[27]. In addition, our IGF1 dose-response experiments show that IGF1R exhibits slightly lower activity in response to very high concentrations of IGF1 (Supplementary Fig. 7).

Different from the active dimer of IGF1R, the two membrane-proximal FnIII-3 domains in the structure of IR ectodomain bound to insulin tightly pack against each other[15]. It remains unclear whether this type of homotypic FnIII-3-FnIII-3′ inter-action is necessary for IR activation. Nevertheless, in this IR conformation, the L1′–FnIII-2′ interaction in the bottom part of the receptor, which is critical for IGF1R activation, seems to become much weaker and is thus unlikely to be functionally important for IR activation. Indeed, the residues of the L1 hairpin involved in the L1′–FnIII-2′ interaction in IGF1R, such as K164 and Y173, are not conserved in IR. It is possible that IGF1R and IR use different strategies to position the two stalk domains near the cell membrane for intracellular kinase auto-phosphorylation.

Collectively, our study provides fresh insight into the mechanism of IGF1R activation and the well-known negative cooperativity in IGF1 binding to IGF1R (Fig. 6), and suggests new strategies for designing antibodies and small molecules that can allosterically regulate IGF1R for the treatment of diseases caused by the dysfunction of IGF1R.

## Methods

**Protein expression and purification**. The MmIGF1R with C terminal tail trun-cation and kinase dead mutation (residues 1–1262, Y951A, D1107N) was sub-cloned into the pEZT-BM expression vector. The mutations were designed based the previous results of the internalization mechanism of IR[12,13]. Based on the high binding affinity of Tsi3 and Tse3 protein, tsi3 gene was fused to the C terminal end of MmIGF1R and the expressed Tsi3 works as the purification tag[28]. HRV-3C protease cleavage sequence was introduced between mIGF1R and Tsi3 genes. MmIGF1R was expressed in HEK293S GnTI- cells (ATCC, CRL-3022) with Bac-to-Bac system. Briefly, the MmIGF1R-3C-Tsi3 fusion construct was transfected into DH10Bac (Thermo Fisher Scientific, 10361012) to produce the recombinant baculovirus. The baculovirus was scaled up by infecting Sf9 insect cells and then it was used to infect HEK293S GnTI⁻ cells at a cell density of $2.5 \times 10^6$ cells per ml with 10 mM sodium butyrate was added to the medium. Infected cells were grown for 48 h at 30 °C before harvesting.

The cells were resuspended in lysis buffer containing 50 mM Tris-HCl pH 8.0, 400 mM NaCl, protease inhibitor cocktail (Roche). The cells were ruptured by french press. 1% DDM was added with stirring for 8 h to solubilize the membrane fraction. The supernatant was isolated by ultracentrifugation for 1 h at 100,000 g.

After 1 mM $CaCl_2$ was added for Tse3-Tsi3 protein interaction, the supernatant was then loaded to the CNBr-activated Sepharose resin (GE Healthcare) conjugated with Tse3 protein by gravity flow. The resin was washed by ten volumes of wash buffer (50 mM Tris-HCl, pH 8.0, 400 mM NaCl, 1 mM $CaCl_2$, 5% glycerol, 0.05% DDM) and mIGF1R protein was eluted by HRV-3C protease cleavage. The MmIGF1R was run on Superose 6 increase 10/300 GL size-exclusion column (GE Healthcare) with buffer containing 20 mM Hepes-Na pH 7.4, 200 mM NaCl, 0.03% DDM to remove most of the misfolded pre-mature mIGF1R. All of the purification step were performed at 4 °C.

Mature human IGF1 gene (residues 49–118) was codon-optimized and inserted into a modified pET-28a vector encoding a SUMO-tag following the His6-tag at the N terminal end. The protein was expressed as inclusion bodies in *E.coli* strain BL21 (DE3) (Thermo Fisher Scientific, C600003). After OD at 600 nm reaches to 0.7, 0.5 mM IPTG was added into the Luria-Bertani Broth medium for 5 h at 37 °C. The inclusion body was dissolved in 8 M urea, 30 mM Tris-HCl pH 8.0, 1 mM EDTA and 5 mM DTT and refolded in buffer containing 0.5 M L-Arginine pH 8.0, 0.6 mM oxidized glutathione overnight at room temperature. The HsIGF1 protein was purified by loading to Ni column (GE Healthcare) and being eluted with ULP1 enzyme cleavage. The protein was further purified with superdex 75 increase 10/300 GL size-exclusion column (GE Healthcare).

To make the complex of MmIGF1R/HsIGF1, purified MmIGF1R and HsIGF1 was mixed at molar ratio 1:2 and incubated for 1 h at 4 °C. The mixture was further purified with SRT500C size-exclusion column (Sepax Technologies) in buffer (20 mM Hepes-Na, pH 7.4, 200 mM NaCl, 0.03% DDM). The peak containing the protein complex was collected and concentrated to 5 mg/ml for cryo-EM analyses.

**EM data acquisition.** The cryo-EM grid was prepared by applying 3 μl of protein samples to glow-discharged Quantifoil R1.2/1.3 300-mesh gold holey carbon grids (Quantifoil, Micro Tools GmbH, Germany). Grids were blotted for 5.0 s under 100% humidity at 4 °C before being plunged into liquid ethane using a Mark IV Vitrobot (FEI). Micrographs were acquired on a Titan Krios microscope (FEI) operated at 300 kV with a K2 Summit direct electron detector (Gatan), using a slit width of 20 eV on a GIF-Quantum energy filter. EPU software (FEI) was used for automated data collection following standard FEI procedures. A calibrated magnification of ×46,730 was used for imaging, yielding a pixel size of 1.07 Å on images. The defocus range was set from −1.5 to −3 μm. Each micrograph was dose-fractionated to 30 frames under a dose rate of 4 e-/pixel/s, with a total exposure time of 15 s, resulting in a total dose of about 50 e−/Å2.

**Image processing.** The detailed image processing statistics are summarized in Supplementary Fig. 2 and 3, and Supplementary Table 1. Motion correction was performed using the MotionCorr2 program[29], and the CTF parameters of the micrographs were estimated using the GCTF program[30]. Initially, ~2000 particles were picked with EMAN2 from a few micrographs[31]. All other steps of image processing were performed using RELION3[32,33]. Class averages representing projections in different orientations selected from the initial 2D classification were used as templates for automatic particle picking from the full datasets. Extracted particles were binned 3 times and subjected to 2D classification. Particles from the classes with fine structural features were selected for 3D classification. Approximately 20,000 particles were selected to generate the initial model in RELION3. Particles from the 3D classes showing good secondary structural features were selected and re-extracted into the original pixel size of 1.07 Å. 3D refinements with no symmetry imposed resulted in 3D reconstructions to 4.8 Å resolution. To improve the resolution, we performed another round of 3D classification by using local search in combination with small angular sampling. The final refinement yielded a map at overall 4.3 Å resolution. All resolutions were estimated by applying a soft mask around the protein density using the gold-standard Fourier shell correlation (FSC) = 0.143 criterion[34]. A portion of the cryo-EM density is shown in Supplementary Fig. 8.

**Model building and refinement.** Model building of the IGF1R-IGF1 complex was initiated by docking each individual domain derived from the crystal structures of IGF1R/IGF1 complex (PDB: 5U8Q) into the cryo-EM map in the program Chimera[35]. The model was manually adjusted in Coot and refined against the map by using the real space refinement module with secondary structure and non-crystallographic symmetry restraints in the Phenix package[36]. The crystal structures of FnIII-3 domains were rigid-body fitted into the cryo-EM density in Chimera, showing good agreement. Model geometries were assessed by using Molprobity as a part of the Phenix validation tools and summarized in Table 1.

**IGF1R activation assay.** 293FT cells (R70007, Invitrogen) were cultured in high-glucose DMEM supplemented with 10% (v/v) FBS, 2 mM L-glutamine, and 1% penicillin/streptomycin. Cells were free from mycoplasma contamination. Plasmid transfection was performed with Lipofectamine™ 2000 (Invitrogen). After 1 day, the cells were serum starved for 16 h and treated with 10 or 50 nM of human IGF1 (100–11, PeproTech) for 5 min. The cells were incubated with the cell lysis buffer [50 mM Hepes pH 7.4, 150 mM NaCl, 10% (v/v) Glycerol, 1% (v/v) Triton X-100, 1 mM EDTA, 100 mM sodium fluoride, 2 mM sodium orthovanadate, 20 mM sodium pyrophosphate, 0.5 mM dithiothreitol (DTT), 2 mM phenylmethylsulfonyl

fluoride (PMSF)] supplemented with cOmplete™ Protease Inhibitor Cocktail (Roche) and PhosSTOP (Roche) on ice for 1 h. After centrifugation at 20,817 g at 4 °C for 20 min, the concentrations of cell lysate were measured using Micro BCA Protein Assay Kit (Thermo Fisher Scientific). Cell lysates (50 μg total proteins) were analyzed by SDS-PAGE and western blotting. Anti-IGF1R-pY1135/1136 (WB: 1:1000; 19H7, Cell signaling; labeled as pY) and anti-MYC (WB: 1:1000; 9E10, Roche; labeled as IGF1R) were used as primary antibodies. For quantitative western blots, anti-rabbit immunoglobulin G (IgG) (H + L) (Dylight 800 conjugates) and anti-mouse IgG (H + L) (Dylight 680 conjugates) (Cell signaling) were used as secondary antibodies. The membranes were scanned with the Odyssey Infrared Imaging System (LI-COR, Lincoln, NE). Levels of pY were normalized to total IGF1R levels and shown as intensities relative to that of IGF1R WT in 50 nM IGF1 treated cells.

**IGF1-binding assay.** Human IGF1 was obtained using the protein refolding method mentioned before. To conjugate IGF1 with Alexa Fluor® 488 (A10235, Thermo Fisher Scientific), human IGF1 in 20 mM Hepes pH 7.5, 100 mM NaCl, and 2 mM EDTA was mixed with Alexa Fluor® 488 NHS Ester dissolved in dimethylsulfoxide (DMSO) at a molar ratio of 1:2 and incubated for 12 h at 4 °C. Labeled IGF1 was separated from the free dye by size exclusion chromatography (Superdex 75 increase 10/300 GL, GE Healthcare) in the buffer containing 20 mM Hepes pH 7.5, 100 mM NaCl, and 2 mM EDTA.

To express IGF1R WT or mutants in 293FT cells, plasmid transfection was performed with Lipofectamine™ 2000 (Invitrogen). After 1 day, the cells were serum starved for 16 h. The cells were incubated with the cell lysis buffer without DTT supplemented with cOmplete™ Protease Inhibitor Cocktail (Roche) and PhosSTOP (Roche) on ice for 1 h. After centrifugation at 20,817 g at 4 °C for 20 min, the concentrations of cell lysate were measured using Micro BCA Protein Assay Kit (Thermo Fisher Scientific). Fifteen microgram of anti-c-MYC magnetic beads (88842, Thermo Fisher scientific) was added to 3 mg of cell lysates and incubated at 4 °C for 2 h. The beads were washed three times with the washing buffer [50 mM Hepes, pH 7.4, 400 mM NaCl, 0.05% NP-40] supplemented with cOmplete™ Protease Inhibitor Cocktail (Roche). The beads were resuspended in 200 μl of binding buffer [20 mM Hepes pH 7.5, 200 mM NaCl, 0.03% DDM, and 0.003% CHS] supplemented with cOmplete™ Protease Inhibitor Cocktail (Roche) and 100 nM of BSA. 20 μl of IGF1R-bound beads and the indicated amount of Alexa Fluor® 488 labeled IGF1 were incubated on a rotator at 4 °C for 1 h. The beads were washed 3 times with the binding buffer. The bound proteins were eluted with 50 μl of binding buffer containing 2% SDS at 50 °C for 10 min. The samples were diluted with 150 μl of binding buffer. The fluorescence intensities were measured in a microplate reader (CLARIOstar; BMG LABTECH). Nonspecific binding was measured in samples of Alexa Fluor® 488 labeled IGF1 with beads without IGF1R and subtracted from the data. The IGF1R amount were analyzed by western blotting (Anti-IGF1R; WB 1:1000; Thermo Fisher Scientific, ZI001).

**In vitro phosphorylation assay.** Hundred nanomolar human IGF1 was added to 5 nM purified mouse IGF1R and incubated on ice for 30 min. Phosphorylation assay was performed in the kinase buffer (0.6% (w/vol) CHAPS, 150 mM NaCl, 25 mM Hepes pH 7.5, 10 mM $MgCl_2$ and 1 mM $MnCl_2$ supplemented with 0.25 mM ATP) and incubated at 25 °C for 5 min. The reaction mixtures were quenched with SDS sample buffer supplemented with 3 mM EDTA and analyzed by western blotting.

**Reporting summary.** Further information on research design is available in the Nature Research Reporting Summary linked to this article.

## Data availability

The cryo-EM map and the fitted coordinates have been deposited in the EMDB and PDB database with accession code EMDB-20524 and PDB: 6PYH. The source data are provided for Figs. 2, 3, 4, 5, and Supplementary Figs. 1, 6, and 7. All other data are available from the corresponding authors on reasonable request.

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

## Acknowledgements

We thank Xuewu Zhang for his comments on our paper. Single particle cryo-EM data were collected at the University of Texas Southwestern Medical Center (UTSW) Cryo-Electron Microscopy Facility that is funded by a Cancer Prevention and Research Institute of Texas (CPRIT) Core Facility Support Award (RP170644). We thank D. Nicastro and D. Stoddard for facility access and data acquisition. X.B. is the Virginia Murchison Linthicum Scholar in Medical Research at UTSW. Research in his lab is supported in part by CPRIT (RR160082) and the Welch foundation (I-1944–20180324). H.Y is an Investigator with the Howard Hughes Medical Institute, and supported by grants from CPRIT (RP120717-P2 and RP160667-P2) and the Welch Foundation (I-1441).

## Author contributions

H.Y. and X.B. conceived the project; J.L. E.C. and H.Y. designed the IGF1R construct for expression; J.L. prepared the IGF1R and IGF1 sample; X.B. performed cryo-EM structure determination and model building; E.C. performed and analyzed cellular and binding assays. All authors participated in paper preparation. H.Y. and X.B. supervised the project.

## Competing interests

The authors declare no competing interests.
