## [Peer Review File · Nature Communications]

Reviewers' comments:

Reviewer #1 (Remarks to the Author):

This study describing the structural basis of the activation of the Type 1 IGF1R, as elucidated by cryo-EM, follows hard on the heels of X-ray crystallographic and biochemical studies of IGF1 binding to IGF1R (refs 7 and 8) and cryo-EM studies of insulin receptor (IR) activation (refs 14 and 16). The present study largely confirms but also extends these earlier studies. The structure is what it is, and aspects relating to the structural basis of ligand binding (including negative co-operativity) and structural transitions associated with activation, are well described.

Structural data

This is not the place for a detailed review of all recent work in this area, but there is scope within the manuscript for clearer analysis of where the present data confirm, extend or call into question conclusions from previous studies of IR and IGFR.

i) The conclusion that the liganded, activated receptor adopts an overall Γ conformation (as compared to the Λ conformation of apo-receptor), reflecting rearrangement of structural domains without significant intra-domain conformational changes, is essentially confirmatory of studies of IR (ref 14 and Gutmann et al, not cited, see below) (line 119). However, it would be interesting to have more comment on just how closely the details of structural transition (lines 128-141) replicate observations made on IR (ref 14).

ii) Likewise the insights into the mechanism of ligand binding and origin of negative co-operativity are acknowledged to be largely confirmatory of previous studies (lines 170 and 301) and again it would be interesting to know whether there are differences in detail compared to other studies, given some inevitable differences reflecting different ligands.

iii) One small but potentially significant difference between this study of IGF1R and a previous study of IR that is briefly highlighted (line 120) concerns the separation distance between FnIII-3 domains (and, by implication, intracellular kinase domains) in activated receptors. However, because both studies used receptor constructs that had been engineered to optimise receptor expression and stability it is impossible to say whether this is a 'real' difference between receptors or a consequence of technical factors.

iv) One aspect of the present study for which novelty is claimed is the definition of an interaction between unliganded L1' and FnIII-2' domains that exists only in the active state and stabilises active dimer (line 250), and of the interface between IGF C-domain loop and receptor CR domain (line 280). In relation to activated IR Wies et al (ref 14) comment only that 'the insulin-free L1' domain retains an apo-like association with domain FnIII-2'' (Fig 2c,d, ref 14). Is the L1'=FnIII-2' interaction unique to the IGF1R, as was it simply unrecognised in IR?

v) A second novel aspect is the definition of the interface between IGF1 C-domain loop and receptor CR domain (line 280), which is proposed to contribute to higher binding affinity of IGF1 compared to IGF2 (which has a smaller C-loop) and insulin (which has no C-loop).

3. Biochemical data

The mutational analyses of secondary binding sites and inter-domain interactions are only weakly supportive of functional significance because they lack controls (for instance mutation of nearby residues not implicated in such contacts) and are at best qualitative (being based on measurements of receptor autophosphorylation at a single time point and with high concentrations of IGF1).

3. Minor points

i) The role of IGF1R, as current understood, is arguably overstated: Abstract line 31 (Dysregulation of IGF1R signalling causes growth retardation and cancer); Introduction line 146 (Dysregulation of IGF1R signalling can lead to a number of human diseases). Would it be more accurate to say that dysregulation of IGF1R has been implicated in human diseases including both growth retardation and cancer?

ii) Gutmann et al (J Cell Biol (2018) 217:1643) were the first to suggest that activated IR adopts a overall Γ conformation, based on single particle negative staining electron microscopy, and their work should be cited.

iii) CRD is undefined as an abbreviation but presumably refers to Cysteine-Rich Domain, in which case it is redundant to refer to the CRD domain

Signed: Kenneth Siddle

Reviewer #2 (Remarks to the Author):

In this manuscript, Li et al. report the cryo-EM structure of IGF1 bound to its receptor, IGF1R, at 4.3-Å resolution. The receptor construct is full-length, including the transmembrane (TM) helices and cytoplasmic tyrosine kinase domains, although neither the cytoplasmic region nor the TM helices is observed in the density (or at least not sufficiently well to model).

The structure is of a 1:1 IGF1:IGF1R complex, which is thought to be the physiologically relevant state for the active receptor. The structure reveals the details of IGF1 binding to domains in both half-receptors, which is the basis for receptor “cross-linking”, similar to what has been observed in cryo-EM structures of the insulin receptor (IR). In addition to the mode of binding of IGF1, the structure shows how this binding mode releases the autoinhibitory constraints (between L1 and FnIII-2') that keep the cytoplasmic kinase domains apart, and how the alphaCT helices, and the disulfide bridge linking these helices, play a structural role in negative cooperativity (in the binding of a second IGF1 molecule). In addition, the structure indicates that, upon binding of one IGF1 molecule to L1 and FnIII-1', the freed (from L1) FnIII-2' makes an intra-protomer interaction with L1', which also maintains its autoinhibitory interaction with FnIII-2. Mutagenesis experiments (of full-length IGF1R in cells) in all of these regions are consistent with the interactions observed in the cryo-EM structure (see point 1 below). This intra-protomer interaction makes an important distinction between this structure of IGF1-IGF1R and a previously reported (Weis et al.) cryo-EM structure of insulin-IR, in which a leucine-zipper was tethered to the C-terminal ends of the ectodomain, resulting in an FnIII-3-FnIII-3' interaction, which is probably artifactual, and an aberrantly small TM-helix separation.

This cryo-EM study, after many structural studies (by cryo-EM and x-ray crystallography) involving either IGF1-IGF1R or insulin-IR, looks to be the first to establish the correct, key interactions that are induced upon 1:1 ligand binding to its receptor, and therefore it represents an important advance in the field.

Points:

1) Although the mutagenesis experiments are consistent with the interactions observed in the cryo-EM structure, all of the experiments are loss-of-function. In particular, the mutagenesis experiments probing the importance of the intra-protomer L1-FnIII-2 interaction for receptor activation could be confounding if the mutations introduced inadvertently affect binding of IGF1 to L1. The authors

could (and should) make a much stronger case for this interaction if they were to design a gain-of-function mutation(s) and show, for example, that the mutant exhibits increased basal-level phosphorylation.

2) In the cartoon in Fig. 6, it would be more realistic if the authors could depict L1' binding to the "bottom" of FnIII-2 (actually, between FnIII-2 and FnIII-3) than to the top of FnIII-2. L1' is not labeled in Fig. 1b, left panel.

Reviewer #3 (Remarks to the Author):

Here Li and colleagues report the cryo-EM structure of the full-length IGF1R receptor in complex with IGF1 at a nominal resolution of 4.8Å of the whole particle, and 4.3Å of the ectodomain. The IGF1R is 57% identical in sequence to the insulin receptor, and plays a role in cell growth. The previously determined X-ray structure of the inactive IGF1R ectodomain informed on auto-inhibition of this receptor tyrosine kinase, however, it remained unclear how ligand (IGF1)-binding relieves this auto-inhibition to form the active IGF1R. In this study, the authors reconstituted the full-length IGF1R-IGF1 complex in detergent micelles, determined its cryo-EM structure, and carried out structure-guided mutational analysis in a cell-based kinase assay. From these results they derive a plausible model for activation of IGF1R that requires only a single IGF1 molecule to bind per IGF1R dimer. The manuscript is clear and well-written and, if the comments below are satisfactorily addressed, is suitable for publication.

Major comments

1. Could IGF1R reconstitution in micelles lead to an artifactual assembly due to the constrained lipid environment, which may bring the IGF1R monomers into closer proximity than in a native lipid-bilayer? Do the authors have evidence that micelle-reconstituted IGF1R has trans-phosphorylation activity, that it is indeed an active state of the receptor?

2. Cryo-EM densities for key regions of the model should be shown clearly in the manuscript (main or ED). This is especially important due to the low resolution of the structure. The density of the IGF1 C-domain in ED Fig. 4 looks of insufficient in quality for modelling side-chains. Please show the density more clearly to convince the reader. If confidence in this region is low, I would suggest the

authors leave this as a polyAla model. The L1' loop shown in Fig. 5b should also be supported by an image of the cryo-EM density.

3. Given the 57% sequence identity to the insulin receptor and the discussion in Lines 118-125, the authors should provide a figure (main or ED) to compare the two (IGF1R and IR) with regard to the ligand binding site and receptor conformation. Similarly, it would be helpful if the authors include a discussion on the similarities or differences between the two in light of the proposed IGF1R activation mechanism.

4. Are wild-type and mutant IGF1R receptors, when co-expressed in HEK cells for the activity assay, expressed to the same levels? Could the authors use two IGF1R variants, one with a GFP or 3xFlag tag attached to blot for the two variants and directly estimate their relative expression levels?

5. The transmembrane and kinase domains of IGF1R remain unresolved, however the authors suggest a crossing of the TM helices of the IGF1R dimers and that this crossing brings the kinase domains into proximity for trans-phosphorylation, see Fig 6 model and ED Fig 4 densities. Can the authors trace density from 1 TM helix to the corresponding IGF1R?

Minor comments

1. Please show the F701 mutant labelled in both IGF1R promoters in e.g. Fig. 2c.

2. Line 97: "97 We were able to unambiguously build a nearly complete model". This is an overstatement given the low resolution and heavy reliance on available crystal structures and the lack of the Kinase domains. Perhaps 'modelled' could be used as a more appropriate term?

3. Many readers will read 'full-length IGF1R' and expect to see the Kinase domains. Can the authors comment on their absence or have they done focused classification in this region of the density?

We thank the reviewers for the positive and constructive comments. In the revised manuscript, we have addressed all the reviewers' concerns by doing additional experiments and by rewriting the manuscript. Doing so has significantly improved our manuscript.

Our point-by-point responses are listed below. For ease of reading, we have colored our responses in blue.

Reviewer #1 (Remarks to the Author):

This study describing the structural basis of the activation of the Type 1 IGF1R, as elucidated by cryo-EM, follows hard on the heels of X-ray crystallographic and biochemical studies of IGF1 binding to IGF1R (refs 7 and 8) and cryo-EM studies of insulin receptor (IR) activation (refs 14 and 16). The present study largely confirms but also extends these earlier studies. The structure is what it is, and aspects relating to the structural basis of ligand binding (including negative co-operativity) and structural transitions associated with activation, are well described.

Structural data

This is not the place for a detailed review of all recent work in this area, but there is scope within the manuscript for clearer analysis of where the present data confirm, extend or call into question conclusions from previous studies of IR and IGF1R.

i) The conclusion that the liganded, activated receptor adopts an overall Γ conformation (as compared to the Λ conformation of apo-receptor), reflecting rearrangement of structural domains without significant intra-domain conformational changes, is essentially confirmatory of studies of IR (ref 14 and Gutmann et al, not cited, see below) (line 119). However, it would be interesting to have more comment on just how closely the details of structural transition (lines 128-141) replicate observations made on IR (ref 14).

We agree with this reviewer (Dr. Siddle) that our Γ -shaped full-length IGF1R-IGF1 complex exhibits an overall shape similar to that of the full-length IR-insulin complex, which is previously studied by negative stain EM (Gutmann et al JCB 2018). It is also similar to the cryo-EM structure of an engineered IR ectodomain bound to insulin, as we discussed in the original submission. We have discussed and cited both papers in the revised manuscript.

We thank Dr. Siddle for the great suggestion of comparing the ligand-induced structural transitions between these two homologous receptors. To do so, we superimposed our model of each IGF1R protomer onto that of the corresponding IR protomer, recently determined by cryo-EM (Weis et. al, Nature Communications 2018; ref 15). This comparison revealed only small structural differences between each related IGF1R and IR protomer, suggesting that highly similar structural transitions occur during the ligand-induced activation of IGF1R and IR. We have added a new supplementary figure (Supplementary Fig. 5b) and a new sentence about the similarity in the IGF1R and IR activation: "Notably, similar structural transitions also occur during the insulin-induced activation of IR (Supplementary Fig. 5b), suggesting that the mechanism underlying activation may be conserved between these two closely related receptors."

ii) Likewise the insights into the mechanism of ligand binding and origin of negative co-operativity are

acknowledged to be largely confirmatory of previous studies (lines 170 and 301) and again it would be interesting to know whether there are differences in detail compared to other studies, given some inevitable differences reflecting different ligands.

As suggested, we have elaborated on the comparisons of the primary and secondary ligand-binding sites of IGF1R and IR in the revised manuscript, and prepared a new figure panel (Fig. 3i) to show the different binding modes. The interaction between the C-domain loop of the ligand and the CRD of the receptor is unique to the IGF1R–IGF1 complex. It represents a major source for ligand specificity between these two homologous, but functional different, receptors.

In terms of negative cooperativity, our structural model and subsequent functional analyses show that the α -CT motifs are the key elements underlying the negative cooperativity in ligand binding to IGF1R. As the α -CT motifs are poorly resolved in all IR structures, we do not know for certain whether the origin of negative cooperativity is the same between IGF1R and IR. Nevertheless, based on the high sequence conservation of the α -CT motif and the previous biochemical result that enhancing the α -CT structural flexibility increases insulin-binding affinity (Weis et. al, Nature Communications 2018), we hypothesize that the origin of negative cooperativity between IR and IGF1R may be similar. We have added this discussion in the revised manuscript.

iii) One small but potentially significant difference between this study of IGF1R and a previous study of IR that is briefly highlighted (line 120) concerns the separation distance between FnIII-3 domains (and, by implication, intracellular kinase domains) in activated receptors. However, because both studies used receptor constructs that had been engineered to optimise receptor expression and stability it is impossible to say whether this is a 'real' difference between receptors or a consequence of technical factors.

We agree with Dr. Siddle that we do not know whether the structural difference observed is a real difference of functional importance. We have rephrased our statement as follows: "Therefore, the structural differences observed here could be due to the different sample preparation methods and may not reflect true differences between the active conformations of these receptors." in the revised manuscript.

iv) One aspect of the present study for which novelty is claimed is the definition of an interaction between unliganded L1' and FnIII-2' domains that exists only in the active state and stabilises active dimer (line 250), and of the interface between IGF C-domain loop and receptor CR domain (line 280). In relation to activated IR Wies et al (ref 14) comment only that 'the insulin-free L1' domain retains an apo-like association with domain FnIII-2'' (Fig 2c,d, ref 14). Is the L1'=FnIII-2' interaction unique to the IGF1R, as was it simply unrecognised in IR?

We thank Dr. Siddle for raising this important question. To address this question, we have carefully compared the ligand-bound conformations of IGF1R and IR. Different from the active dimer of IGF1R, the two membrane-proximal FnIII-3 domains in the structure of IR ectodomain bound to insulin tightly pack against each other. In this IR conformation, the L1'-FnIII-2' interaction in the bottom part of the receptor, which is critical for IGF1R activation, seems to become much weaker and is thus unlikely to be functionally important for IR activation. Consistent with this observation, the residues of the L1 hairpin involved in the L1'-FnIII-2' interaction in IGF1R, such as K164 and Y173, are not conserved in IR.

Therefore, it is possible that IGF1R and IR use different strategies to position the two stalk domains near the cell membrane for intracellular kinase auto-phosphorylation. Further structural and biochemical studies will be required to fully reveal the similarities and differences in IGF1R and IR activation. We have added a new paragraph and a new supplementary figure (Supplementary Fig. 5a, b) in the revised manuscript.

v) A second novel aspect is the definition of the interface between IGF1 C-domain loop and receptor CR domain (line 280), which is proposed to contribute to higher binding affinity of IGF1 compared to IGF2 (which has a smaller C-loop) and insulin (which has no C-loop).

We thank Dr. Siddle for recognizing this important novelty of our study. As discussed above, this interaction is a major contributor to ligand specificity.

3. Biochemical data

The mutational analyses of secondary binding sites and inter-domain interactions are only weakly supportive of functional significance because they lack controls (for instance mutation of nearby residues not implicated in such contacts) and at best qualitative (being based on measurements of receptor autophosphorylation at a single time point and with high concentrations of IGF1).

As suggested, we mutated two residues (N529 and D531), which were located at the periphery of the secondary binding site, as negative controls. These two mutations have no effect on IGF1R activation (Fig. 3d, e). We also introduced more point mutations at the L1 hairpin. Mutations of the residues with large charged or aromatic side chains, such as K164A and Y171A, cause deficiency in IGF1-dependent activation. In contrast, mutations of residues with small and uncharged side chains, such as T166A or N169A, have no effect on IGF1R activation (Fig. 5e, f). To further exclude the possibility that these L1 hairpin mutations can affect ligand binding, we performed *in vitro* IGF1-binding assays, and showed that IGF1R WT and 163GS (with the entire L1 hairpin replaced with a Gly-Ser linker) can bind IGF1 with similar affinities (Fig. 5g, h). We have incorporated these new results in the revised figures, and described these new results in the revised manuscript.

As suggested by the reviewer by Dr. Siddle, we performed dose and time response activity assays for IGF1R WT, R488E and 163GS. Both R488E and 163GS mutants exhibited reduced IGF1-dependent IGF1R activation in a wide range of IGF1 concentrations and at different time points. These new results are included in a new supplementary figure (Supplementary Fig. 6).

3. Minor points

i) The role of IGF1R, as current understood, is arguably overstated: Abstract line 31 (Dysregulation of IGF1R signalling causes growth retardation and cancer); Introduction line 146 (Dysregulation of IGF1R signalling can lead to a number of human diseases). Would it be more accurate to say that dysregulation of IGF1R has been implicated in human diseases including both growth retardation and cancer?

Point accepted. We have revised our writing accordingly.

ii) Gutmann et al (J Cell Biol (2018) 217:1643) were the first to suggest that activated IR adopts a overall Γ conformation, based on single particle negative staining electron microscopy, and their work should be cited.

We apologize for missing this citation. We have cited this important work in the revised manuscript.

iii) CRD is undefined as an abbreviation but presumably refers to Cysteine-Rich Domain, in which case it is redundant to refer to the CRD domain

We have defined CRD and changed the term "CRD domain" to simply "CRD" in the revised manuscript.

Reviewer #2 (Remarks to the Author):

In this manuscript, Li et al. report the cryo-EM structure of IGF1 bound to its receptor, IGF1R, at 4.3-Å resolution. The receptor construct is full-length, including the transmembrane (TM) helices and cytoplasmic tyrosine kinase domains, although neither the cytoplasmic region nor the TM helices is observed in the density (or at least not sufficiently well to model).

The structure is of a 1:1 IGF1:IGF1R complex, which is thought to be the physiologically relevant state for the active receptor. The structure reveals the details of IGF1 binding to domains in both half-receptors, which is the basis for receptor "cross-linking", similar to what has been observed in cryo-EM structures of the insulin receptor (IR). In addition to the mode of binding of IGF1, the structure shows how this binding mode releases the autoinhibitory constraints (between L1 and FnIII-2') that keep the cytoplasmic kinase domains apart, and how the alphaCT helices, and the disulfide bridge linking these helices, play a structural role in negative cooperativity (in the binding of a second IGF1 molecule). In addition, the structure indicates that, upon binding of one IGF1 molecule to L1 and FnIII-1', the freed (from L1) FnIII-2' makes an intra-protomer interaction with L1', which also maintains its autoinhibitory interaction with FnIII-2. Mutagenesis experiments (of full-length IGF1R in cells) in all of these regions are consistent with the interactions observed in the cryo-EM structure (see point 1 below). This intra-protomer interaction makes an important distinction between this structure of IGF1-IGF1R and a previously reported (Weis et al.) cryo-EM structure of insulin-IR, in which a leucine-zipper was tethered to the C-terminal ends of the ectodomain, resulting in an FnIII-3-FnIII-3' interaction, which is probably artifactual, and an aberrantly small TM-helix separation.

This cryo-EM study, after many structural studies (by cryo-EM and x-ray crystallography) involving either IGF1-IGF1R or insulin-IR, looks to be the first to establish the correct, key interactions that are induced upon 1:1 ligand binding to its receptor, and therefore it represents an important advance in the field.

This is a very clear and succinct summary of our work. We thank the reviewer for the positive comments.

Points:

1) Although the mutagenesis experiments are consistent with the interactions observed in the cryo-EM structure, all of the experiments are loss-of-function. In particular, the mutagenesis experiments probing

the importance of the intra-protomer L1-FnIII-2 interaction for receptor activation could be confounding if the mutations introduced inadvertently affect binding of IGF1 to L1. The authors could (and should) make a much stronger case for this interaction if they were to design a gain-of-function mutation(s) and show, for example, that the mutant exhibits increased basal-level phosphorylation.

We agree with the reviewer that gain-of-function mutants are extremely powerful for validating the functional importance of this newly found L1-FnIII-2 interaction. Structure-based design of gain-of-function mutations, however, is very difficult. In addition, our cryo-EM map of the IGF1R–IGF1 complex determined at moderate resolution does not reveal the atomic details of this L1-FnIII-2 interaction, which makes the design of gain-of-function mutations even more challenging. Despite of this technical difficulty, we still attempted to do this and designed a series of mutations of the L1 hairpin, including T166D, T166Y, N169F, and N169L, with the goal of increasing the affinity of the L1-FnIII-2 interaction. Because the residues from the FnIII-2 domain involved in this interaction are mainly hydrophobic, we hoped that changing the interacting residues of the L1 hairpin from hydrophilic residues to hydrophobic ones might have increased the affinity between these two motifs. Unfortunately, none of our designed IGF1R mutants showed higher basal-level activities, and most had decreased IGF1-induced activities, as compared with IGF1R WT (see the figure below). Thus, these mutations may weaken the L1-FnIII-2 interaction, rather than strengthening it. We speculate that the hydrophobic residues introduced at the L1 hairpin may not be compatible with those in the FnIII-2 domain, causing unintended clashes. We hope that the reviewer would understand the difficulty of designing gain-of-function mutants.

Figure-for-Reviewer | The effect of mutations at L1 hairpin.

To help ease this reviewer’s concerns, we performed *in vitro* IGF1-binding assays. IGF1R WT and 163GS (one of mutations disrupting the L1-FnIII-2 interaction) bound to IGF1 with similar affinities, which at least excluded the possibility that these L1 hairpin mutations affected IGF1 binding. We have incorporated these new results in Fig. 5g, h, and described them in the revised manuscript.

2) In the cartoon in Fig. 6, it would be more realistic if the authors could depict L1’ binding to the “bottom” of FnIII-2 (actually, between FnIII-2 and FnIII-3) than to the top of FnIII-2. L1’ is not labeled in Fig. 1b, left panel.

We thank the reviewer for this great suggestion. We have revised Fig. 1,4 and 6 as suggested.

Reviewer #3 (Remarks to the Author):

Here Li and colleagues report the cryo-EM structure of the full-length IGF1R receptor in complex with IGF1 at a nominal resolution of 4.8Å of the whole particle, and 4.3Å of the ectodomain. The IGF1R is 57% identical in sequence to the insulin receptor, and plays a role in cell growth. The previously determined X-ray structure of the inactive IGF1R ectodomain informed on auto-inhibition of this receptor tyrosine kinase, however, it remained unclear how ligand (IGF1)-binding relieves this auto-inhibition to form the active IGF1R. In this study, the authors reconstituted the full-length IGF1R-IGF1 complex in detergent micelles, determined its cryo-EM structure, and carried out structure-guided mutational analysis in a cell-based kinase assay. From these results they derive a plausible model for activation of IGF1R that requires only a single IGF1 molecule to bind per IGF1R dimer. The manuscript is clear and well-written and, if the comments below are satisfactorily addressed, is suitable for publication.

Major comments

1. Could IGF1R reconstitution in micelles lead to an artifactual assembly due to the constrained lipid environment, which may bring the IGF1R monomers into closer proximity than in a native lipid-bilayer? Do the authors have evidence that micelle-reconstituted IGF1R has trans-phosphorylation activity, that it is indeed an active state of the receptor?

We thank the reviewer for raising this good point. We agree that the detergent micelle might affect the receptor conformation, which could potentially lead to receptor inactivation. To ease his/her concern, we performed *in vitro* phosphorylation assays using the recombinant IGF1R isolated with the detergent. The IGF1R with detergent showed IGF1-induced activation in the presence of ATP, although it exhibited high background activity even in the absence of IGF1. We reason that the high basal-level activity is caused by the cross-phosphorylation between two IGF1R dimers by three-dimensional diffusion-collision in the buffer solution. When IGF1R is inserted into the cell membrane, the orientation of the kinase domain might be restrained, such that this type of inter-dimer cross-phosphorylation may be prevented. We have prepared a new figure panel (Supplementary Fig. 1c, d) to show this result and discussed it in the revised manuscript.

2. Cryo-EM densities for key regions of the model should be shown clearly in the manuscript (main or ED). This is especially important due to the low resolution of the structure. The density of the IGF1 C-domain in ED Fig. 4 looks of insufficient quality for modelling side-chains. Please show the density more clearly to convince the reader. If confidence in this region is low, I would suggest the authors leave this as a polyAla model. The L1' loop shown in Fig. 5b should also be supported by an image of the cryo-EM density.

Point accepted. We have remade Supplementary Fig. 4 to clearly show the cryo-EM density of IGF1 in different orientations. We agree with the reviewer that the cryo-EM density of the IGF1 C-domain loop is insufficient for modelling the side chains. Following this reviewer's comment, we have removed all the side chains from the model of the C-domain loop, with the exception of Y31, whose side chain is clearly resolved in our cryo-EM map and can thus be built confidently. We also prepared a new figure panel to show the cryo-EM density of the L1 loop in Supplementary Fig. 4.

3. Given the 57% sequence identity to the insulin receptor and the discussion in Lines 118-125, the authors should provide a figure (main or ED) to compare the two (IGF1R and IR) with regard to the ligand binding site and receptor conformation. Similarly, it would be helpful if the authors include a discussion on the similarities or differences between the two in light of the proposed IGF1R activation mechanism.

We thank the reviewer for these good suggestions. We have prepared several new figures to compare the ligand-binding sites and receptor conformations of IGF1R and IR (Fig. 3i, Supplementary Fig. 5). We have also elaborated on the similarities or differences of the ligand-induced activation mechanisms between these two homologous receptors in the revised manuscript.

4. Are wild-type and mutant IGF1R receptors, when co-expressed in HEK cells for the activity assay, expressed to the same levels? Could the authors use two IGF1R variants, one with a GFP or 3xFlag tag attached to blot for the two variants and directly estimate their relative expression levels?

We thank the reviewer for raising this good point. To determine the relative expression of IGF1R WT and F701A in cells, we transfected the 1:1 mixture of Myc-tagged IGF1R WT and untagged IGF1R F701A into cells. The Western blot result showed that IGF1R WT and F701A were expressed at similar levels. We have prepared a new figure panel to show this new result (Fig. 2f, g).

5. The transmembrane and kinase domains of IGF1R remain unresolved, however the authors suggest a crossing of the TM helices of the IGF1R dimers and that this crossing brings the kinase domains into proximity for trans-phosphorylation, see Fig 6 model and ED Fig 4 densities. Can the authors trace density from 1 TM helix to the corresponding IGF1R?

Based on our current cryo-EM map, we cannot trace the density from one TM to the corresponding FnIII-3 domain of IGF1R. Nevertheless, as there are only 6 residues connecting the C-terminus of FnIII-3 to the N-terminus of TM, we can confidently assign each TM helix to the adjacent FnIII-3 domain.

Minor comments

1. Please show the F701 mutant labelled in both IGF1R promoters in e.g. Fig. 2c.

Point accepted. We have made the change accordingly.

2. Line 97: "97 We were able to unambiguously build a nearly complete model". This is an overstatement given the low resolution and heavy reliance on available crystal structures and the lack of the Kinase domains. Perhaps 'modelled' could be used as a more appropriate term?

Point accepted. We have toned down the statement based on the reviewer's suggestion.

3. Many readers will read 'full-length IGF1R' and expect to see the Kinase domains. Can the authors comment on their absence or have they done focused classification in this region of the density?

We thank the reviewer for raising this good point. As suggested by this reviewer, we performed focused classification with density subtraction (Bai *et. al*, 2015 eLife), a method described in our previous paper. In this approach, the density corresponding to the extracellular domain of IGF1R is removed from each individual particles images, resulting a new particle set with only TM and intracellular kinase domains. We then performed focused refinement on the intracellular kinase domain. Despite these efforts, resolution for the intracellular domain was not improved, probably because the kinase domain is small and featureless. In addition, we cannot exclude the possibility that a portion of our protein complexes, particularly the kinase domain, may be damaged by the hydrophobic air-water interface on the cryo-EM grid. To improve the resolution of TM and intracellular kinase domains, we as a field need to develop better sample preparation methods for this type of single-span transmembrane receptors, so that the orientation of the intracellular kinase domain can be fixed with respect to the extracellular domain.

In conclusion, we are grateful to the referees for their thoughtful and critical comments that have greatly improved our manuscript.

REVIEWERS' COMMENTS:

Reviewer #1 (Remarks to the Author):

The authors have responded very fully and positively to all the comments of the original reviewers. They have added some additional data, new figures, and discussion, that have significantly improved what was already an interesting and significant study. I am content for the revised version to be accepted for publication.

Reviewer #2 (Remarks to the Author):

The authors have satisfactorily addressed my concerns.

Reviewer #3 (Remarks to the Author):

The authors have addressed all my comments satisfactorily, I am happy to support this manuscript for publication.

REVIEWERS' COMMENTS:

Reviewer #1 (Remarks to the Author):

The authors have responded very fully and positively to all the comments of the original reviewers. They have added some additional data, new figures, and discussion, that have significantly improved what was already an interesting and significant study. I am content for the revised version to be accepted for publication.

Thanks this reviewer for the positive comments.

Reviewer #2 (Remarks to the Author):

The authors have satisfactorily addressed my concerns.

Thanks this reviewer for the positive comments.

Reviewer #3 (Remarks to the Author):

The authors have addressed all my comments satisfactorily, I am happy to support this manuscript for publication.

Thanks this reviewer for the positive comments.